# Local Ablative Therapy Associated with Immunotherapy in Locally Advanced Pancreatic Cancer: A Solution to Overcome the Double Trouble?—A Comprehensive Review

**DOI:** 10.3390/jcm11071948

**Published:** 2022-03-31

**Authors:** Jonathan Garnier, Olivier Turrini, Anne-Sophie Chretien, Daniel Olive

**Affiliations:** 1Departement of Surgical Oncology, Institut Paoli-Calmettes, 232 Boulevard de Sainte Marguerite, 13009 Marseille, France; turrinio@ipc.unicancer.fr; 2Centre de Recherche en Cancérologie de Marseille, Aix-Marseille University, 58 Boulevard Charles Livon, 13007 Marseille, France; anne-sophie.chretien@inserm.fr (A.-S.C.); daniel.olive@inserm.fr (D.O.); 3Team Immunity and Cancer, U1068 Inserm, UMR7258 Centre National de la Recherche Scientifique, 13009 Marseille, France; 4Departement of Immunomonitoring, Institut Paoli-Calmettes, 232 Boulevard de Sainte Marguerite, 13009 Marseille, France

**Keywords:** locally advanced pancreatic cancer, irreversible electroporation, immunotherapy, immune checkpoint inhibitor

## Abstract

Pancreatic ductal adenocarcinoma (PDAC) remains a major killer and is a challenging clinical research issue with abysmal survival due to unsatisfactory therapeutic efficacy. Two major issues thwart the treatment of locally advanced nonresectable pancreatic cancer (LAPC): high micrometastasis rate and surgical inaccessibility. Local ablative therapies induce a systemic antitumor response (i.e., abscopal effect) in addition to local effects. Thus, the incorporation of additional therapies could be key to improving immunotherapy’s clinical efficacy. In this systematic review, we explore recent applications of local ablative therapies combined with immunotherapy to overcome immune resistance in PDAC and discuss future perspectives and challenges. Particularly, we describe four chemoradiation studies and nine reports on irreversible electroporation (IRE). Clinically, IRE is the ablative therapy of choice, utilized in all but two clinical trials, and may create a favorable microenvironment for immunotherapy. Various immunotherapies have been used in combination with IRE, such as NK cell- or γδ T cell-based therapy, as well as immune checkpoint inhibitors. The results of the clinical trials presented in this review and the advancement potential of these therapies to phase II/III trials remain unknown. A multiple treatment approach involving chemotherapy, local ablation, and immunotherapy holds promise in overcoming the double trouble of LAPC.

## 1. Introduction

Pancreatic ductal adenocarcinoma (PDAC), the pancreatic cancer with the highest incidence, remains a major killer and is one of the most challenging clinical research issues. The 5-year survival rate in PDAC is one of the poorest among cancers, with only 10% in 2021 [1], due to insufficient therapeutic efficacy and is projected to increase, with PDAC expected to be the second most lethal cancer by 2030 [2]. The combination of neoadjuvant chemotherapy and surgical resection proves to be optimal only for a few patients (15–20%) with resectable tumor at diagnosis. Thus, the vast majority of patients present with either locally advanced pancreatic cancer (LAPC) characterized by extensive vascular involvement (30–40%) or metastatic disease (40–50%) [3]. Cytotoxic chemotherapy is the main treatment for advanced disease; however, in PDAC, this approach is only marginally effective, limited by its immunosuppressive effects and cumulative toxicity. During the last decade, several advancements have been made in the treatment of LAPC.

A key innovation in this direction was the introduction of FOLFIRINOX (a drug combination comprising leucovorin, 5-fluorouracil, irinotecan, and oxaliplatin), which has improved the median overall survival (OS) of LAPC patients from 9 to 19 months [4,5] and which has been shown to attenuate LAPC to a resectable disease in 10–60% of the patients [6,7,8]. Although the benefit of surgery after FOLFIRINOX needs to be confirmed by randomized trials, numerous studies have reported a survival rate of 22–39 months from diagnosis in patients undergoing surgical resection after FOLFIRINOX therapy [9,10]. Moreover, a combination of gemcitabine and nab-paclitaxel is an alternative in first line therapy that seems to be well tolerated [11,12]. However, the assessment of resectability after induction chemotherapy by imaging is often difficult [13], and computed tomography (CT) cannot accurately differentiate the edges of viable tumor tissue. Owing to this uncertainty in determining resectability, surgical exploration could be useful for patients without progression according to the response evaluation criteria in solid tumors (RECIST). Nevertheless, many patients might still be unresectable during surgical exploration due to either local extension or metastasis. Accordingly, there are two key issues (or a double trouble) with PDAC management: resistant micrometastasis and local extension.

The first issue is that PDAC is a resistant micrometastatic disease, even for resectable tumors as defined by the borderline biological classification [14,15], highlighting the need for physicians to look beyond the anatomical difficulties related to the location of the pancreas and to propose systemic therapy as the cornerstone of the treatment. Micrometastasis requires improved diagnostic and therapeutic strategies. First, during diagnosis, patients tend to be falsely diagnosed to have resectable disease because of the missed recognition of distant metastases rather than the underestimated major vascular invasion, leading to futile surgery in up to 19% [16] of the cases. Biological or radiological surrogate markers could help predict the futility of surgery [17]. Missed recognition of distant metastasis could be primarily attributed to metastatic lesions that are too small to be detected by any currently available imaging modalities. Although metabolic imaging could predict early recurrence [18], novel biomarkers are urgently needed [19]. Second, mechanisms of resistance to chemotherapy for PDAC remain unclear but multifactorial, and the resistance could either be an initial property of the tumor or be acquired upon drug exposure. This resistance is presumed to arise from the interactions among the tumor microenvironment (TME) [20,21], pancreatic stem cells [22], and cancer cells [23]. Another important factor is hypoxia [24], which induces a desmoplastic reaction in the TME and leads to increased immune and chemo-resistance. Third, even if immunotherapy is a recent breakthrough, showing impressive responses in melanoma [25,26] and non-small-cell lung cancer [27,28], it is not equally promising to be used for PDAC treatment. Recently, Ma et al. have reported a significantly longer survival time in patients treated with a combination of chemotherapy and checkpoint inhibitors than in patients treated with chemotherapy alone in advanced pancreatic cancer treatment (18.1 vs. 6.1 months) [29], without the use of a FOLFIRINOX-based regimen. The combination of gemcitabine with nab-paclitaxel and immunotherapy is being evaluated in various ongoing registered clinical trials (two terminated or completed and nine recruiting, as per the clinicaltrial.gov platform), and so is FOLFIRINOX therapy (two completed and four recruiting). Three major hurdles for the comprehensive implementation of immunotherapy in PDAC have been identified: a low mutational load [30] associated with the downregulation of major histocompatibility complex (MHC)-I, an unfavorable local immunologic TME [31,32,33,34,35,36,37], and infiltrating T cells with profound alterations [38]. In order to attack cancer cells, the immune system needs a recognition process initiated by antigen-presenting cells (APCs) that present tumor antigens on the MHC molecules on cell surfaces. This specific trigger leads to the expression of co-stimulatory molecules on APCs and enhances lymph node migration, where the antigen is then presented to T cells via the antigen-specific T cell receptor (TCR). If the co-stimulatory molecules interact with the ligands on T cells, activation begins, and the T cells leave the lymph node. When the activated T cells recognize tumor antigens, the release of cytokines and cytolytic enzymes induces proliferation, resulting in tumor lysis and T cell memory acquisition. Disruption of any step of this process leads to the immune evasion of tumors. Moreover, PDAC is generally associated with a characteristic dense desmoplastic stroma comprising cancer-associated fibroblasts, pancreatic stellate cells, and extracellular matrix, resulting in hypoxic and hypovascular tumors [23,39]. This powerful immunosuppressive TME is presumed to exclude or restrict T cell access to tumor cells, a phenomenon known as an immune privilege [40], in which the tumor is protected from immune attack. Moreover, both CTLA-4 (circulating CD4^+^ and CD8^+^ T cells) and PD-L1 expression upregulation are associated with worse survival [41,42,43,44]. Furthermore, gamma delta T (γδT) cells, which constitute up to 40% of the tumor infiltrating lymphocytes in PDAC, express high levels of PD-L1, contributing to the suppression of T cell activation, as determined by preclinical data [45]. Finally, preliminary trials of checkpoint inhibitors in pancreatic cancer have yielded disappointing results for ipilimumab (anti-CTLA-4) [46] and anti-PD-L1 antibodies [25,47], and PDAC is known as the “dark side” of immunotherapy [48]. Thus, a reaction catalyst is needed to initiate a response.

The second problem is local extension, as the pancreas is situated between the celiac trunk, hepatic artery, and superior mesenteric vessels, rendering explicit its surgical inaccessibility. Thermal ablation (radiofrequency ablation (RFA) or microwave ablation) is associated with high morbidity when applied to pancreatic tumors due to the presence of these fragile structures [49,50]. Thus, irreversible electroporation (IRE) [51] and other non-thermal methods (e.g., immunostimulating interstitial laser thermotherapy [52] or stereotactic body radiation therapy (SBRT) [53]) are emerging tools for the management of patients with locally advanced disease. RFA, IRE, and SBRT are all feasible techniques for treating LAPC, with acceptable morbidity rates and a median OS of 23 months [49].

There is increasing evidence that in addition to the local effect, these local ablative therapies can also induce a systemic antitumor response with a potential abscopal effect [54,55,56]. The incorporation of additional therapies able to “start the fire” and convert the “cold” microenvironment to a “hot” microenvironment (i.e., with T cell infiltration and production of proinflammatory cytokine) could become a pivotal strategy to enhance the clinical efficacy of immunotherapy [57]. Thus, this synergistic combination is presumed to overcome the double trouble of first, an inextirpable tumor, by proposing a local ablation, and second, a micro-metastatic disease, by reprogramming the immune system to chase non-visible tumoral cells at any location in the body. Figure 1 summarizes the current treatment strategies for LAPC, emphasizing the notion that using chemotherapy as the cornerstone for treatment appears reasonable.

This systematic review aimed to report preclinical and clinical studies of local ablative therapies combined with immunotherapy that have recently been proposed to overcome the “double trouble” with advanced PDAC.

## 2. Methods

The Preferred Reporting Items for Systematic Reviews and Meta-Analyses (PRISMA) 2009 guidelines [58] were used as a guide for this review although it is not a systematic review.

### 2.1. Study Selection

A search of PubMed (United States National Library of Medicine, http://www.ncbi.nlm.nih.gov/pubmed, accessed on 1 December 2021), Scopus (Elsevier, http://www.scopus.com/, accessed on 1 December 2021), Web of Science (Thomson Reuters, http://apps.webofknowledge.com/, accessed on 1 December 2021), and Google Scholar (https://scholar.google.it/, accessed on 1 December 2021), for studies published in English was performed. Different domains of medical subject heading (MeSH) terms and keywords were combined with ‘AND,’ and terms within domains were combined with ‘OR.’ The first domain related to locally advanced pancreatic cancer, and the second contained terms related to treatments or ablative therapies. Terms were restricted to the title, abstract, and keywords. A full description of the search strategy is available in Appendix A. We selected the studies published before the end of November 2021. Duplicates were removed, and articles were screened by the title and abstract for eligibility independently by two authors (J.G. and O.T.).

### 2.2. Eligibility Criteria

The inclusion criteria were studies that specified LAPC according to the National Comprehensive Cancer Network (NCCN) guidelines [59] or the American Joint Committee on Cancer 8th edition [60] or described the tumor anatomy, which was congruent with these criteria. According to the NCCN criteria, LAPC is defined as a pancreatic adenocarcinoma without overt distant metastases and is associated with >180° involvement of the hepatic artery, superior mesenteric artery, and/or celiac trunk, or unreconstructible involvement of the porto-mesenteric vein.

Studies were included if they were related to ablative therapies associated with immunotherapy in patients with LAPC and reported data for one or more of the following endpoints: immune effect, morbidity, mortality, and/or survival. The exclusion criteria were (a) review articles, (b) case reports (or studies with a population of less than 10), (c) studies with only combined results for stage III–IV disease or including borderline resectable with an inability to derive results, (d) primary-recurrent disease, and (e) published only in abstract form. Figure 2 provides a flow diagram for the literature search.

### 2.3. Statistical Analysis

Outcomes are expressed as they were originally reported. A meta-analysis was not performed owing to the obvious heterogeneity between studies. Statistical analyses were not performed owing to the variation in the outcomes reported and techniques.

## 3. Results

### 3.1. Preclinical Settings

Direct damage to the tumor results in the release of tumor-derived peptides for processing by APCs and subsequent T cell priming. In addition, cancer cell apoptosis leads to the production of several factors recognized by APCs, such as dendritic cells (DCs). This release of damage-associated molecular patterns or DAMPs (endogenous molecules released upon stress or by dying cells carrying a danger signal) activates DCs and promotes antitumor immunity by priming naïve cytotoxic T cells via MHCII presentation.

Radiation therapy, a common approach, has recently been shown to modify the immunosuppressive TME of PDAC [61]. RFA, even if reported in cases of unresectable PDAC, is limited by thermal injury to adjacent tissues [62]. With respect to RFA for LAPC, Giardino et al. [63] showed that the levels of CD4^+^, CD8^+^, and effector memory T cells increased from day 3 and later those of myeloid antigen-presenting dendritic cells increased on day 30, suggesting an adaptive immune response. After RFA, circulating IL-6 increased dramatically on day 3 and decreased to baseline by day 30, consistent with the supposed antitumor effect. Alternatively, IRE is a new non-thermal and minimally invasive technology that can cause tumor cell death by short high-voltage electric pulses. IRE modulates the peritumoral stroma and enriches the antitumor immune response. Inducing the permanent porosity of the tumor stroma causes a transient decrease in three of the most clinically correlated Treg populations (CD4^+^CD25^+^, CD4^+^CD25^+^FoxP3^+^, and CD4^+^CD25^+^FoxP3^−^) from 3 to 5 postoperative days after in situ IRE of stage III PDAC [64], and exhibiting a stronger abscopal effect than that of thermal ablation [65].

With respect to in vivo studies, there are two reports of chemoradiation (including one comparing with IRE), one with SBRT and three with invasive treatments, all involving IRE associated with immunotherapy, with no reports of RFA, microwave ablation, or cryotherapy ablation. The results of these studies are summarized in Table 1, except those of the study by Mills et al. [66], on SRBT and local interleukin-12 microsphere (IL-12 MS), which is not considered a “classical” immunotherapy but could be interesting.

Mills et al. demonstrated an increase of intratumoral interferon gamma production following SBRT/IL-12 MS administration, which initiates suppressor cell reprogramming, and a subsequent increase in CD8 T cell activation. Furthermore, activating systemic tumor immunity is capable of eliminating established liver metastases, resulting in marked tumor reduction and cure in multiple preclinical mouse models of PDAC. However, chemoradiation is limited by the radio-resistance of PDAC and negative results of phase III trials [67,68]. Moreover, Zhao et al. found that the combination of IRE and checkpoint inhibition was more effective than irradiation and the same checkpoint inhibitor [56]. They showed that combined IRE and anti-PD1 immune checkpoint blockade significantly suppressed tumor growth and prolonged the survival of immunocompetent mice bearing well-established orthotopic PDAC or melanoma tumors. Remarkably, they showed that the survivors rejected tumor cell rechallenge, along with an anti-tumor memory T cell response, and demonstrated that tumor-infiltrating CD8^+^ T cells are a key contributor to the superior anti-tumor efficacy of IRE + anti-PD1.

Apart from conventional immunotherapy, oncolytic viruses contribute to a promising category of immunotherapy drugs that can activate both innate and adaptative responses [69]. Sun et al. [70] reported that a combination therapy of IRE and M1 oncolytic virus enhanced local and systemic T cell activation in the zinc-associated protein deficient situation, which was estimated to occur in 84% of the patients in their study. Other studies used checkpoint inhibitors in combination. Azad et al. [71] found that PD-L1 blockade can improve PDAC response to chemoradiation and can “shift” the balance toward a more favorable immune phenotype, providing an important insight into the potential of immune checkpoint inhibitors to radiosensitize PDAC. Narayan et al. [72] showed that combining IRE with intratumoral Toll-like receptor-7 (TLR7) agonist (1V270) and PD-1 blockade improved treatment responses and eliminated untreated concomitant distant tumors (abscopal effects), an effect which was not observed with IRE alone. These results suggest that the systemic anti-tumor immune response triggered by IRE can be enhanced by stimulating the innate immune system. This is highlighted by the findings of O’Neil et al. [73] and Sun et al. [70], who showed that PD-L1 expression is induced in pancreatic cancer after IRE.

Finally, similar to IRE, advanced radiation techniques such as SBRT are challenging to model in mice, and direct comparisons in humans will be necessary.

**Table 1 jcm-11-01948-t001:** Summary of in vitro and in vivo studies.

Reference	Local Therapy	Immunotherapy	Taxon	Key Results
Azad et al. [71]	Radiation	Anti-PD-L1	Mice	Radiosensitizing with higher RT doses Reduces infiltration of CD11b + Gr1 + myeloid cells Enhances infiltration of CD45 + CD8 + T cells
Narayanan et al. [72]	IRE	Anti-PD-1 + TLR7 agonist	Mice	Requires an intact immune system Induces a systematic adaptative response Prophylactic immunity to tumor rechallenge Increased CD8 + DCs
Zhao et al. [56]	IRE vs. radiation	Anti-PD-1	Mice	IRE is superior to radiation Suppresses tumor growth 35–43% Prophylactic immunity to tumor rechallenge Key contributor: TI CD8 + T cells
O’Neil et al. [73]	IRE	Anti-PD-L1	Mice	Increases PD-L1 expression
Sun et al. [70]	IRE	M1 Oncolytic Virus	Mice	Combination improved anticancer efficacy Prolonged survival orthotopic animals Enhance local and systemic T cells activation

IRE: Irreversible Electroporation; TI: Tumor-infiltrating.

### 3.2. Clinical Studies

Concerning local ablative therapies for PDAC, published studies of combined treatment with immunotherapy are scarce. While high-intensity focused ultrasound has not been considered in the present review, owing to the limited data on its oncological outcomes, this may serve as a promising procedure for pain relief [74]. Studies of microwave, cryoablation, or their combination with immunotherapy are lacking. Concerning radiotherapy, several ongoing trials have not yet been published. RFA has an immune effect in liver tumors [75], and its combination with a CTLA-4 inhibitor had a clinical benefit in a phase 1 trial of hepatocellular carcinoma [76]. To date, there are no published studies of unresectable PDAC. The ablative therapy of choice is IRE, as it has been applied in all clinical trials identified in this review. Several recent studies have confirmed that IRE could be a promising procedure for treating LAPC, while ensuring the preservation of the blood vessels and nerves, and may offer a survival benefit in unresectable PDAC [77,78]. IRE has been combined with various immunotherapies, as described below and summarized in Table 2.

Of note, cetuximab (mAb anti-Epidermal Growth Factor Recepto) [79] and oregovomab (mAb anti CA 125) [80] treatments are not stricto sensu considered as immunotherapy and were excluded in this review, but these treatments could provide interesting data in this narrow field.

Except patients in the Pan [81] study who were not pre-treated, half of the patients underwent previous chemotherapy treatment with gemcitabine [82,83] and nab-paclitaxel [73,83,84] or FOLFIRINOX [73,82,83,84] in various ratios.

**Table 2 jcm-11-01948-t002:** Clinical phase I/II studies.

Reference	Local Therapy	Systemic Therapy	Nb of Patients	Survival (Median DFS/OS *)	Key Messages
Lin et al. [82]	IRE	Allogeneic NK cells	35	9.1/13.6	Potentially synergistic Augmented OS
Pan et al. [81]	IRE	Allogeneic NK cells	92	7.2/12.4	Radiological and biological
O’Neil et al. [73]	IRE	Anti-PD-L1	10	6.8/18	Feasible Effector memory T cells
Lin et al. [83]	IRE	Allogeneic Vγ9Vδ2 T cells	62	18.5/22.5	Augmented OS and PFS Improves QoL
He et al. [84]	IRE	Anti-PD1	15	23.4/44.3	Potentially synergistic Augmented OS and PFS

* in months; RT: Radiotherapy; SBRT: stereotactic body radiation therapy; NK: natural killer; QoL: quality of life.

#### 3.2.1. Natural Killer (NK) Cells

For decades, NK cells have been known as broadly “nonspecific” killer cells, different from cytotoxic T lymphocytes or other immunocytes that identify specific targets. NK cells are trained to recognize “non-self” histocompatibility antigens (human leukocyte antigen, HLA) on the surface of cells via immunoglobulin-like receptors. Recent insights into their ability to produce immune-active cytokines have made them particularly attractive tools for immunotherapy. In view of this, in 2017, Lin et al. [82] brought NK cells into clinical research in a trial involving patients with unresectable pancreatic cancer. They have shown that this therapy is efficient and feasible, with limited adverse events (grade I and II). The median progression-free survival (PFS) after IRE was higher in IRE-NK than in IRE alone (9.1 vs. 7.9 months, *p* = 0.0432), as was the median OS (13.6 vs. 12.2 months; *p* = 0.0327). Moreover, the response was dose-dependent, as the median PFS after IRE for patients who received multiple NK infusions was higher than that of patients who only received a single NK infusion (9.9 vs. 8.2 months; *p* = 0.0387). In 2020, Pan et al. [81] confirmed the safety and showed that IRE + NK is associated with a better objective response (with an improved objective response rate from 56.5% to 71.7%; *p* = 0.038) and biological response (serum CA 19–9 level at 30 days post-treatment: 359 versus 475 KU/L; *p* = 0.019) compared to those for IRE alone, but does not impact survival. Among immunological indicators, the IRE-NK group had remarkably higher levels of serum IL-2, TNF-β, and IFN-γ than those in the IRE group after treatment (*p* < 0.05), while no statistically significant differences were observed in the levels of IL-4, IL-6, and IL-10 between the two groups (*p* > 0.05).

#### 3.2.2. Immune Checkpoint Inhibitors

To date, there are only two published trials with checkpoint inhibitors [73,84], with one trial involving a total of 25 patients with unresectable PDAC. The combination of IRE and the anti-PLD1 [73] antibody nivolumab was given successfully without any dose reduction (nivolumab 240 mg). However, 70% of patients experienced grade ≥3 treatment-related adverse events (40% were attributable to IRE and 20% to nivolumab). Survival was not compared. The other study [84] used the anti-PD1 Toripalimab (240 mg) with acceptable toxic effect and found a benefit compared with IRE alone on PFS (27.5 vs. 10.6 months, *p* = 0.036) and OS (44.3 vs. 23.4, *p* = 0.010). One of the study aims from O’Neil et al. [73] was to evaluate the establishment of memory T cells after nivolumab treatment associated with IRE. Three populations of CD4^+^ memory T cells were evaluated (CD3^+^, CD4^+^, CD45, and CCR7^+^ naïve T cells; CD3^+^, CD4^+^, CCR7^+^, CD45RA^−^ central memory cells, and CD3^+^, CD4^+^, CCR7^−^CD45^−^ effector memory T cells) in the peripheral blood. The effector memory cells increased from baseline nearly 2-fold by postoperative day 90 (*p* = 0.009). There were no differences for CD4^+^ T cells, naïve T cells, or central memory T cells. Of note, the statistically significant differences occurred at 3 months postoperatively, following the completion of immunotherapy. In a study by He et al. [84], the number of CD4^+^ (*p* = 0.038) and CD8^+^ (*p* = 0.024) T cells increased while that of CD8^+^ Treg cells (*p* = 0.023) decreased in the IRE plus toripalimab treatment group compared with those in the IRE-only treatment group.

#### 3.2.3. Vγ9Vδ2 T Cells

In the TME, T cells are a key component of antitumor immune surveillance, and the enhancement or restoration of their antitumor function is the main objective of immunotherapies based on immune checkpoint inhibitors or adoptive cell infusion. Most clinical applications of T cells are centered on αβ T cells (CD4^+^, CD8^+^ T cells); however, γδ T cells also have important roles in cancer immunity [85]. γδ T cells constitute 0.5–16% of the total CD3^+^ cells in the peripheral blood and can be activated in an MHC-independent manner, providing relatively broad antitumor cytotoxicity. Thus, Vγ9Vδ2 T cells are a major component that contribute to immune surveillance against many types of tumors [86,87] indicating that they are a promising candidate for PDAC treatment. However, to date, only one study of cholangiocarcinoma [88] has demonstrated that allogenic γδ T cell treatment could improve immune functions, depletes tumor activity, upgrade the quality of life, and extend survival. In their study, Lin et al. [83] randomly assigned 62 patients to groups treated by IRE or IRE combined with allogeneic Vγ9Vδ2-T cell infusion. Sixty percent of patients were pretreated with FOLFIRINOX. Major (grade ≥ 3) adverse effects occurred in 22% of patients and were related to IRE, with no difference between the two groups. Tumor growth was significantly lower for the combination treatment (*p* < 0.005) than in the group treated by IRE alone. The combination of IRE and allogenic Vγ9Vδ2 T cells enhanced immune function. In addition to an increase in the absolute lymphocyte count, the levels of IL-2, IFN-γ, and TNF-β were also improved, suggesting a robust antitumor efficacy [89]. The combination of IRE and allogeneic Vγ9Vδ2 T cells not only enhanced the γδ T cell effect but also inhibited PD-1 expression. Compared with the results of the IRE-only treatment, the levels of CA19-9 and those of circulating tumor cells decreased following the combination treatment. Furthermore, the combination of IRE and allogeneic Vγ9Vδ2 T cells yielded substantial improvements in median PFS and OS, which were significantly different from those in the IRE only group, even when considering the time from treatment (PFS 11 vs. 8.5 months, *p* = 0.03 and OS 14.5 vs. 11 months, *p* = 0.01). Multiple ɣδ T cell infusions further improved OS but not PFS compared with those for single ɣδ T cell infusions.

#### 3.2.4. Future Directions

The increasing attention received by the research community suggests immunotherapy as a promising option for a multitude of malignancies. We are just beginning to understand both tumor and patient-specific factors determining the efficacy of the treatment options. Although the response to single-agent immune stimulation (whether by radiotherapy, local ablative therapies, or immune checkpoint inhibitors) is weaker in pancreatic cancer than in other more immunogenic cancers, the immune system can still play a role in treatment. Notably, several stages of treatments may be needed, including therapies that counter the immunosuppressive microenvironment, increase antigen presentation, and prime the immune system. The vast majority of currently open trials seek to stimulate an adaptive immune response in pancreatic cancer and aim to target multiple immune response factors, often with a combination of the standard of care therapy (surgery, chemotherapy, radiotherapy) and either immune checkpoint inhibitors or other immune-priming agents. We are awaiting the results of the clinical trials presented in this review and the advancement of these therapies to phase II/III trials, hoping that outcomes can be improved for this recalcitrant disease. However, data are still scarce, as only 13 trials were identified for IRE, 8 for RFA (mainly endoscopic and for neuroendocrine tumors), 23 for SBRT, 2 for microwave and ImLT, and 55 for immunotherapy in PDAC. Table 3 provides an overview of ongoing trials of local ablative therapy associated with immunotherapy for advanced PDAC, and Figure 3 summarizes the potential objectives of combinations of local ablative therapy and immunotherapy.

## 4. Conclusions

The overwhelming majority of patients with pancreatic cancer have no indications for surgical resection; therefore, chemotherapy, radiation, local ablative treatments, or other innovative treatments are essential. Triplet chemotherapy by FOLFIRINOX is currently the gold standard of care and should be investigated further, as this review focused on marginal treatments after induction chemotherapy. Immunotherapy is a major area of recent research aimed at overcoming cancer drug resistance and is of particular interest in treating PDAC. Modalities are not yet established, and checkpoint inhibitors, oncolytic viruses, and immune cell infusion (NK cells or γδ T cells) are of major interest. However, other treatments, such as stroma-targeted therapy, vaccination cancer therapy, and multiple combinations, should be investigated. From a clinical perspective, among local ablative therapies, IRE is the most promising with a more efficient local ablative capacity than those of other physical processes and a more substantial abscopal effect. Our review was limited by the very narrow scope and specific topic (i.e., LAPC with ablative therapy combined with immunotherapy) and the scarce data available in the literature. However, the effectiveness of combined therapeutic approaches for LAPC is a promising area of research. Expectedly, a single therapeutic approach does not appear to be suitable for all patients; therefore, it is necessary to develop biomarkers [90] to tailor treatments to the individual, as the immune landscape of pancreatic cancer has emerged as an important prognostic feature [23,91]. These results pave the way toward the clinical translation of combinatorial treatment strategies that capitalize on the ability to create an immunostimulatory microenvironment, and thinking outside of the box is essential to obtain feasible combinations.

Ultimately, a clinical strategy combining local ablation with therapy that enhances both the adaptive and innate immune systems will likely be necessary to prevent progression and recurrence in this highly refractory disease.

## Figures and Tables

**Figure 1 jcm-11-01948-f001:**
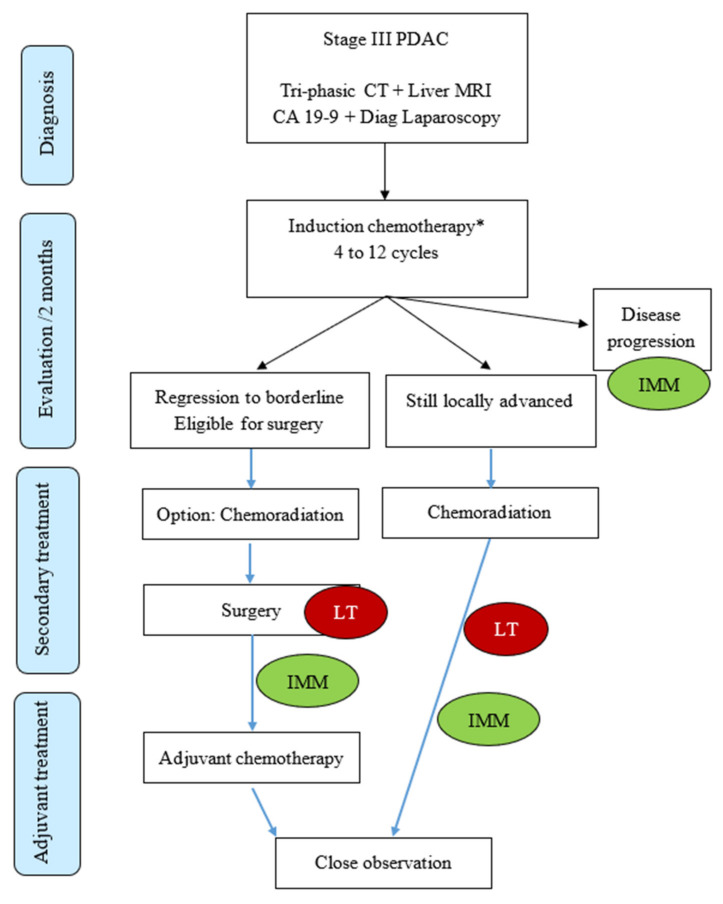
Current therapeutic strategies for locally advanced PDAC. Red and green bubbles indicate potential targets for local therapy (LT) and immunotherapy (IMM), respectively. * FOLFIRINOX or Gemcitabine with Nab-Paclitaxel; CT: computed tomodensitometry; MRI: Magnetic Resonance Imaging; CA 19-9: Carbohydrate Antigen 19-9.

**Figure 2 jcm-11-01948-f002:**
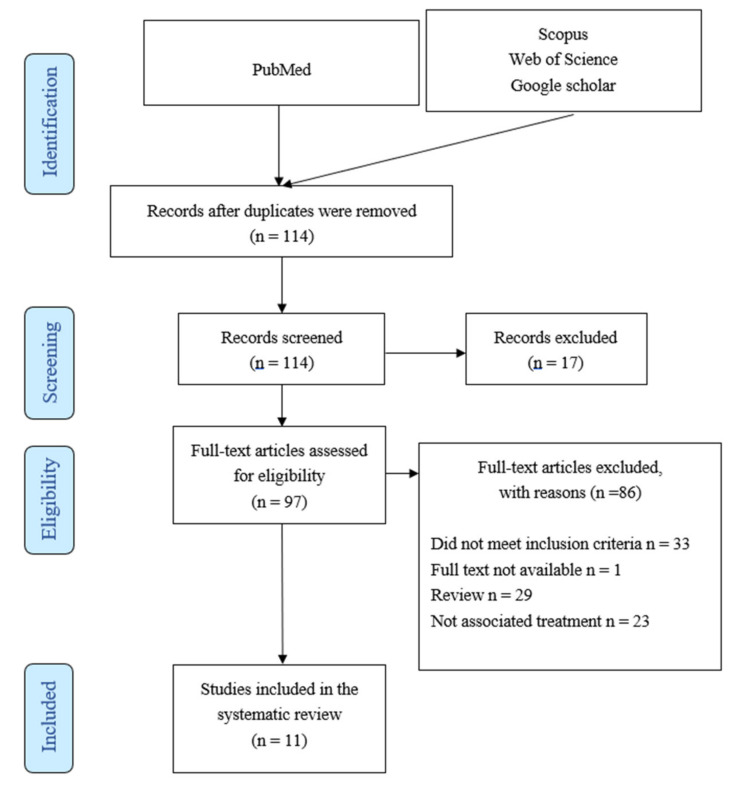
PRISMA Flow Diagram.

**Figure 3 jcm-11-01948-f003:**
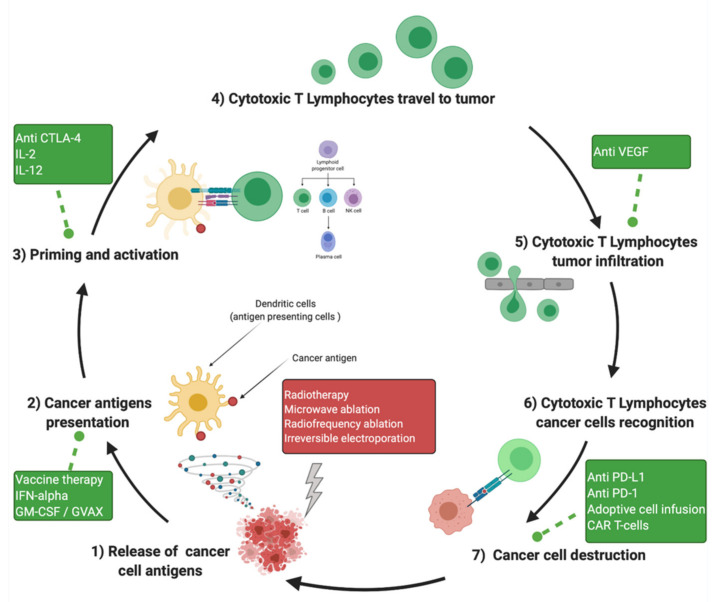
Cancer immunity cycle and immunotherapy intervention at different steps of the antitumor response. Created with BioRender.com accessed on 20 December 2021. GM-CSF: Granulocyte-Macrophage Colony Stimulating Factor.

**Table 3 jcm-11-01948-t003:** Ongoing trials involving combined treatment strategies for advanced PDAC (stage III/IV).

NCT Number	Physical Process	Immunotherapy	Number of Patients	Allocation	Completion Date
03778879	SBRT	CCX872-B			withdrawn
03716596	SBRT	Anti-PD1	36	Single Arm	October 2021
02866383	Radiation	Nivol/Ipilimumab	160	Randomized	November 2021
03563248	SBRT	Nivolumab	160	Randomized	December 2021
02648282	SBRT	Pembrolizumab + GVAX	54	Single Arm	January 2022
03767582	SBRT	Nivolumab + GVAX	30	Randomized	March 2022
04612530	IRE	Nivolumab + TLR9	18	Randomized	October 2022
04098432	SBRT	Nivolumab	20	Single Arm	December 2022
04156087	MWA	Durvalumab + Tremelimumab	20	Single Arm	December 2023
03080974	IRE	Nivolumab	10	Single Arm	June 2025

SBRT: Stereotactic body radiation therapy; TLR9: Toll-like receptor 9; MWA: Microwave ablation.

## Data Availability

Not applicable.

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
