# Peer review of "Local Ablative Therapy Associated with Immunotherapy in Locally Advanced Pancreatic Cancer: A Solution to Overcome the Double Trouble?—A Comprehensive Review"

_jcm, 2022, doi:10.3390/jcm11071948_

Round 1

Reviewer 1 Report

The Authors present an interesting and well executed systematic review exploring the current evidence with regard to local ablative therapies (including IRE) combined with immunotherapy for locally advanced PDAC. Present challenges and possible future perspectives are also extensively discussed.

The review is overall well written, and presents a comprehensive overview on an interesting and developing topic. Despite germinal, immunotherapy may represent an important tool in the future multimodality treatment of PDAC, and local therapies may overcome some of its current limitations. Moreover, the authors correctly underline that systemic chemotherapy (with either FOLFIRINOX or Gem-Abraxane) is the current standard and should remain the cornerstone of the treatment algorithm.

I have no major comment. I would suggest a scientific english review checking for minor defects.

Reviewer 2 Report

In this review entitled “ Local ablative therapy associated with immunotherapy in locally advanced pancreatic cancer: A solution to overcome the double trouble? – A systematic review”, Garnier et al address the issue of the problematic treatment of LAPC, in particular the combination strategy between local therapy and immunotherapy.

It is a very specific but interesting field, and the major criticism of the review is the limited data and results available; but it is appreciable that, in the last section of the draft, authors declare this problem. Because of this, in general, I think that the systematic review is not the best format and that a simple review format could help to make this limit less relevant.

In order to be suitable for publication, the original draft should be subjected to some major revisions.

Specific comments:

  1. In the title and in the introduction section, authors introduce this concept of double trouble in pancreatic cancer. I think that it is confusing and not well discussed. In fact, in the introduction section, despite they state that the two issues are the concept of “micrometastatic disease” and the local extension of PDAC, they argue several critical aspects among the most important in the context of this review is the resistance to immunotherapy and the concept of “cold” TME. Authors could refine this point in order to render more incisive the draft.

  1. Moreover, the important rationale on which it is based this combination strategy, is more discussed in the preclinical settings I think that this part could be moved in the introduction and replaced by more informations about the preclinical studies included in Table 1.

  1. I think that this is not recommended to include in clinical studies of Table 2, studies referred to anti EGFR and anti CA-125 therapies, because they are not strictly considered immunotherapy, in fact authors have decided to not discuss them in a sub paragraph.

  1. There is not correspondence between the number of clinical trials mentioned in Future directions section, line 332-333 (“only 13 trials were identified for IRE, eight for RFA, 23 for SBRT, two for microwave and ImLT, and 55 for immunotherapy in PDAC”) and the 10 trials included in Table 3.

Round 2

Reviewer 2 Report

  • Because of this, in general, I think that the systematic review is not the best format and that a simple review format could help to make this limit less relevant.

Response: Thank you for bringing our attention to this point. As suggested, we have modified the review type, from “systematic review” to a more generalized “comprehensive review”.

Response: In my opinion, it is not sufficient to change the title; authors would to adapt the format of the review to the new “comprehensive review” format, so eliminating the PRISMA Flow Diagram and all the Methods section, related to the quality assessment. In fact, these aspects are not requested in this type of review.

  • Moreover, the important rationale on which it is based this combination strategy, is more discussed in the preclinical settingsI think that this part could be moved in the introduction and replaced by more informations about the preclinical studies included in Table 1.

Response: As suggested, we have revised the Introduction section by introducing the rationale of the study and have deleted certain portions to avoid redundancy (the changes are highlighted in yellow). However, we did not add more information about the preclinical studies, given the medical audience of the journal, to present a “digest” type manuscript, as we have already discussed the translational concepts extensively in the clinical studies part. However, if you feel that the results from the preclinical studies should be discussed more and included in the manuscript to improve understanding, we will be happy to insert them.

Response: I think that it can be exhaustive to summarize results in the table and discuss in the section only the more important studies. But, again, I see a discrepancy between the text (“With respect to 232 in vivo studies, there are two reports of chemoradiation, one with SBRT and three with 233 invasive treatments, all involving IRE associated with immunotherapy, with no reports of 234 RFA, microwave ablation, or cryotherapy ablation”), and the five preclinical studies reported in Table 1.

For the other comments, I think that the changes in the text are in line with the requests.
